# *C1orf106* (*INAVA*) Is a SMAD3-Dependent TGF-β Target Gene That Promotes Clonogenicity and Correlates with Poor Prognosis in Breast Cancer

**DOI:** 10.3390/cells13181530

**Published:** 2024-09-12

**Authors:** Lauren S. Strathearn, Lindsay C. Spender, Christina Schoenherr, Susan Mason, Ruaridh Edwards, Karen Blyth, Gareth J. Inman

**Affiliations:** 1Division of Cellular Medicine, School of Medicine, University of Dundee, Dundee DD1 4HN, UK; lauren.strathearn@fccc.edu (L.S.S.); l.spender@dundee.ac.uk (L.C.S.);; 2Cancer Research UK Scotland Institute, Garscube Estate, Bearsden, Glasgow G61 1BD, UK; c.schoenherr@beatson.gla.ac.uk (C.S.); s.mason@beatson.gla.ac.uk (S.M.); karen.blyth@glasgow.ac.uk (K.B.); 3Division of Molecular and Clinical Medicine, School of Medicine, University of Dundee, Dundee DD1 4HN, UK; 4Institute of Cancer Sciences, University of Glasgow, Glasgow G61 1BD, UK

**Keywords:** C1orf106, INAVA, TGF-β, breast carcinoma

## Abstract

Transforming Growth Factor-β (TGF-β) can have both tumour-promoting and tumour-suppressing activity in breast cancer. Elucidating the key downstream mediators of pro-tumorigenic TGF-β signalling in this context could potentially give rise to new therapeutic opportunities and/or identify biomarkers for anti-TGF-β directed therapy. Here, we identify *C1orf106* (also known as innate immunity activator *INAVA*) as a novel TGF-β target gene which is induced in a SMAD3-dependent but SMAD2/SMAD4-independent manner in human and murine cell lines. C1orf106 expression positively correlates with tumourigenic or metastatic potential in human and murine breast cancer cell line models, respectively, and is required for enhanced migration and invasion in response to TGF-β stimulation. C1orf106 promoted self-renewal and colony formation in vitro and may promote tumour-initiating frequency in vivo. High *C1orf106* mRNA expression correlates with markers of aggressiveness and poor prognosis in human breast cancer. Taken together, our findings indicate that C1orf106 may act as a tumour promoter in breast cancer.

## 1. Introduction

Transforming growth factor-β (TGF-β) signalling influences autonomous tumour cell and tumour microenvironment processes to paradoxically suppress or promote tumourigenesis dependent upon cell context [1,2,3,4]. Through activation of cell surface Type I and Type II serine/threonine kinase receptors, TGF-β stimulates a signalling cascade involving phosphorylation of the receptors SMADs, SMAD2 and SMAD3, which then form a complex with the common mediator SMAD4. In a dynamic process, the SMAD complexes translocate to the nucleus, where alongside other DNA-binding transcription factors and co-activators and co-repressors, they positively and negatively regulate the transcription of TGF-β target genes [5,6]. TGF-β signalling can activate and crosstalk with several non-canonical pathways to influence gene expression and cell behaviour, which include ERK/p38/JNK MAPKs, PI3K/AKT and RHO GTPases [7].

Mutation of core TGF-β pathway signalling effectors occurs in several malignancies, which can allow tumours to evade the tumour-suppressive actions of TGF-β at the early stages of tumourigenesis. Conversely, in other tumour types such as those arising in the breast, functional canonical signalling is retained but tumour-suppressor functions are disabled via epigenetic and genetic mechanisms [8,9]. The dual role of TGF-β in tumourigenesis is well established in breast cancer. Genetic manipulation of TGF-β ligands or receptors in oncogenic murine mammary carcinoma models has demonstrated a growth inhibitory role for TGF-β signalling in the early stages of tumourigenesis but enhanced metastatic progression in advanced disease [10,11,12,13,14]. Similarly, cell line models of breast cancer progression implicate TGF-β as a tumour suppressor in normal and pre-malignant cells and as a pro-metastatic factor as cells gain tumourigenicity [15,16,17]. In further support of TGF-β driving tumour progression, high levels of TGF-β1 ligand are expressed at regions of invasion and metastasis in clinical breast tumour specimens [18,19], and mRNA expression of a TGF-β response signature in patients is associated with increased incidence of breast cancer metastasis [16].

Understanding the specific contexts in which TGF-β either suppresses or promotes tumourigenesis is of crucial importance and could result in the identification of suitable downstream mediators of TGF-β that could be targeted therapeutically. Here, we identify *C1orf106* as a novel TGF-β target gene and investigate its roles in cancer. Genome-wide association studies (GWAS) linked single nucleotide polymorphisms in the *C1orf106* gene (also known as innate immunity activator *INAVA*) to increased risk of inflammatory bowel disease (IBD) [20,21]. Functionally, low *C1orf106* expression in models of IBD contributes to the disease phenotype by impairing inflammatory cytokine production [22] and epithelial barrier permeability [23,24]. Further, it is amplified in breast cancer, maintaining the basal-like/luminal progenitor appearance [25]. In this work, we examine its potential role in TGF-β-induced tumour promotion in breast cancer.

## 2. Materials and Methods

### 2.1. Cell Lines and Culture Conditions

The cell lines used were kind gifts from Caroline Hill (Francis Crick Institute, London, UK): A431 (human epidermoid carcinoma), MI, MII, MII and MIV (MCF10A metastatic progression series of human breast carcinoma) [17]; Lalage Wakefield (Center for Cancer Research, National Cancer Institute, Rockville, MD, USA): 67NR (mouse mammary tumour) and 4T07 (mouse mammary tumour); and Peter ten Dijke (Leiden University, The Netherlands): mouse embryonic fibroblasts (MEFs; wild type, SMAD2 −/−, SMAD3 −/− and SMAD4 −/−). A549 (human lung carcinoma) cells were purchased from the American Type Culture Collection (ATCC, LGC Standards, Teddington, UK); AKR-2B (mouse fibroblasts) and HepG2 (human hepatocellular carcinoma) cells were purchased from the European Collection of Authenticated Cell Cultures (ECACC, Salisbury, UK); HEK293T (human embryonic kidney) cells were obtained from Dharmacon (Lafayette, CO, USA) and 4T1-luciferase (4T1-luc; mouse mammary tumour) cells were purchased from Sibtech Inc. (Brookfield, CT, USA). Cells were routinely tested for mycoplasma contamination. A431, A549, HepG2, AKR-2B, MEFs, HEK293T, MI, MII, MII, MIV and 4T1-luc cells were maintained in DMEM supplemented with 10% fetal bovine serum (FBS) and 2 mM glutamine. The 67NR and 4T07 cells were grown in RPMI-1640 supplemented with 10% FBS and 2 mM glutamine. TGF-β, Activin, BMP2/4/6/7/9 and 10 were obtained from Peprotech (London, UK), and the TGF-β receptor 1 (TGFBR1) inhibitor SB-431542 was purchased from Tocris (Bristol, UK).

### 2.2. Plasmids, siRNAs and Generation of Stable Cell Lines

To generate stable *C1orf106* knockdown cell lines, lentiviral-based pGIPZ shRNA (Horizon Discovery, Cambridge, UK) constructs were used. HEK293T cells were transfected with shRNA plasmids in combination with viral packaging plasmids psPAX2 and pMD2.G. Target cells were incubated in virally conditioned medium, and transfection efficiency was assessed by GFP imaging. Next, transduction cells were selected in puromycin, generating pools of *C1orf106* knockdown cells. Two individual shRNA constructs were used to knockdown murine *C1orf106* (5730559C18Rik) and human *C1orf106*: Human *C1orf106*: shRNA 1 (#RHS4430-200297041) and shRNA 2 (RHS-4430200288118); murine *C1orf106*: shRNA 1 (#RMM4431-200412401) and shRNA 2 (#RMM4431-200348404). A non-silencing pGIPZ construct (#RHS4346) was used as a negative control.

For the transient knockdown of *Smad3* and *Smad4* two individual OnTarget Plus siRNAs or a non-targeting control (Horizon Discovery; #J-040706 (*Smad3*); #J040687 (*Smad4*); #D-001810 (non-targeting control)) were used to transfect *Smad2* −/− MEFs with Lipofectamine RNAi (Life Technologies, Renfrew, UK).

For overexpression studies, cDNA encoding full-length *C1orf106* (NM_001142569.2) was cloned into pcDNA5/TO/FRT C-GFP or pCMV5D C-Flag by the MRC PPU cloning service, University of Dundee. *C1orf106*-tag cDNA was then subcloned into pcDNA4/TO mammalian expression vectors (Invitrogen, Renfrew, UK) using MluI and BamHI restriction sites. A *C1orf106* N-terminal deletion mutant lacking sequence encoding amino acids 2–131 (NM_001142569.2) in the pcDNA5/TO/FRT C-GFP vector was made by the MRC-PPU and was subsequently subcloned into pcDNA4/TO, as described above.

### 2.3. Measurement of Gene and Protein Expression

For mRNA expression analysis, cells were lysed in Tri Reagent (Thermo Fisher Scientific, Renfrew, UK), and phenol:chloroform extraction was performed to isolate RNA, which was then quantified and checked for RNA integrity. cDNA was generated by reverse transcription (DyNAmo cDNA synthesis kit; Thermo Fisher Scientific, Renfrew, UK), and gene expression was analysed by quantitative real-time PCR (qRT-PCR) using the DyNAmo SYBR green qPCR kit (Thermo Fisher Scientific, Renfrew, UK) according to the manufacturer’s instructions. Quantitect primers (Qiagen, Manchester, UK) were used to amplify human *C1orf106* (#QT00086268); human *HPRT1* (#QT00059066); human *18s* ribosomal RNA (#QT00199367); murine *C1orf106* (#QT00255969) and murine *Hprt1* (#QT00166768). *C1orf106* Ct values were normalised to either *HPRT1* or *18s*, and data represent 2^ΔCt^.

For analysis of protein expression, cells were harvested in 4× SDS sample buffer (4% sodium dodecyl sulphate; 0.1 M Tris-HCl pH 6.8; 20% glycerol; 2% *v*/*v* 2-mercaptoethanol) or RIPA buffer containing a protease inhibitor cocktail (Roche, Welwyn Garden City, UK). Where necessary, protein extracts were quantified using the Bio-Rad DC Assay (Bio-Rad, Watford, UK) according to the manufacturer’s instructions. Lysates were analysed by SDS-PAGE and Western blotting using the following antibodies: C1orf106 (rabbit polyclonal, #HPA027499, Merck, Gillingham, UK; 1:1000); C1orf106 (mouse monoclonal, clone OTI6A6, #TA807187, Origene, Herford, Germany; 1:1000); phospho-SMAD2 S465/467 (rabbit monoclonal, clone 138D4, #3108, Cell Signaling Technology, Leiden, The Netherlands; 1:500); phospho-SMAD3 S423/425 (rabbit monoclonal, clone EP823Y, #ab52903, Abcam, Cambridge, UK; 1:2000), β-actin (mouse monoclonal, clone AC-74, #A2228, Merck, Gillingham, UK; 1:10,000) and α-tubulin (Merck, Gillingham, UK; 1:10,000). The bound primary antibody was detected with secondary HRP-conjugated antibodies (DAKO, Prickwillow, UK) and enhanced chemiluminescence (GE Healthcare, Chalfont St Giles, UK). Densitometry measurements were determined in ImageJ and C1orf106 expression was normalised to β-actin or α-tubulin for each sample.

### 2.4. Migration and Invasion Assays

Cell migration was assessed by scratch wound assay using the Woundmaker™ and IncuCyte^©^Zoom imaging platforms (Essen Biosciences, Royston, UK). Cells were seeded to confluency in 96-well microtiter plates and then serum-starved overnight in 0.1% FBS/DMEM. Uniform wounds were generated across each cell monolayer and cells were imaged every two hours to determine wound confluency. Treatments were added immediately post-scratch. Time-lapse videos allowed tracking of cells to confirm migration was occurring as opposed to proliferation.

The xCELLigence RTCA DP system (ACEA Biosciences, San Diego, CA, USA) was used to measure cell invasion in real-time. Cells were detached using DMEM supplemented with 200 mM EDTA, and 4 × 10^4^ were seeded in DMEM (0.1% FBS) in the upper chamber of a CIM-Plate 16 (Roche, Welwyn Garden City, UK) on top of a layer of 20% reduced growth factor Matrigel. DMEM with 2% FBS was added to the lower chamber to act as a chemoattractant. Cell index (indicator of invasion) was measured by the xCELLigence system every 15 min for up to 48 h.

### 2.5. Colony Formation and Clonogenicity Assays

For mammosphere formation assays, cells were seeded at 5000–8000 cells per well in 6-well plates coated with poly 2-hydroxyethyl methacrylate (poly-hema) to prevent adhesion. After five days in culture, colonies > 50 μm in size were counted by light microscopy using a 10× objective and eyepiece graticule. To assess adherent colony formation, cells were seeded at 100–250 cells per 10 cm tissue culture dish. After 7–14 days, colonies were fixed and stained with a 1:1 mix of borax and toluidine blue and colonies >50 cells were counted. To measure clonogenic potential, cells were seeded at 1 cell per well in a 96-well plate and incubated for 7–14 days before fixation in 100% methanol and staining colonies with 0.4% *w*/*v* sulphorhodamine blue in 1% acetic acid (SRB). Colonies with >50 cells were counted, and the plating efficiency and surviving fraction of cells were calculated [26].

### 2.6. Mammary Fat Pad Tumour Initiation

Animal studies were carried out under the Animal Scientific Procedures Act 1986 and EU Directive 2010/63/EU in accordance with UK Home Office regulations (Project licence 70/8645 held by K Blyth), adhering to the ARRIVE guidelines, and were reviewed and approved by the Animal Welfare and Ethical Review Board of the University of Glasgow. 67NR cells containing empty vector or C1orf106-GFP in a 1:1 Matrigel:PBS mixture (50 μL) were orthotopically transplanted at limiting dilution (100, 50 and 10 cells) into the 4th mammary fat pad of 8-week-old CD1-Nude female mice (Charles River, Tranent, UK) surgically under anaesthesia and with analgesia. There were 10 mice per group. Animals were housed in groups of 5 in filter-top cages in a 12 h/12 h light/dark cycle with environmental enrichment and ad libitum access to food and water. Tumour size was measured thrice weekly by calliper measurement by technicians blinded to the outcome of the study. Tumours were allowed to reach a maximum dimension of 15 mm^3^ in any direction, and overall tumour volume was calculated using the formula volume = (length × width^2^)/2.

### 2.7. Survival Analysis of Clinical Datasets

Meta-analysis of unselected breast cancer patient survival in relation to *C1orf106* mRNA expression (probe ID: 219010_at) was carried out using KMPlot (http://kmplot.com/analysis/index.php; accessed on 2 February 2023) [27]. Patient groups showing high and low *C1orf106* expression were determined from median *C1orf106* mRNA expression levels, and the indicated survival endpoint was shown. *C1orf106* mRNA expression according to breast cancer molecular subtype was analysed in Breast Cancer Gene Expression Miner v4.1 using default settings under the “expression” module [28].

### 2.8. Statistical Analysis

When comparing two individual groups, an unpaired student’s *t*-test was performed to determine the statistical significance between the means of the two groups of data. More than two groups were compared by one-way ANOVA followed by post-hoc Dunnett’s or Tukey’s test. Log rank survival analysis was employed to assess significant differences in in vivo data.

## 3. Results

### 3.1. C1orf106 Is a TGF-β Target Gene in Human and Mouse Cells

*C1orf106* was identified by exon array as a gene upregulated by TGF-β stimulation of A431 vulval squamous cell carcinoma cells. We validated *C1orf106* as a transcriptional target gene of TGF-β using qRT-PCR; there was a significant increase in *C1orf106* mRNA expression induced by stimulation of A431 cells with exogenous TGF-β (Figure 1a). Increased expression of C1orf106 protein following TGF-β treatment of A431 cells was confirmed by Western blotting (Figure 1b). Basal expression in this cell line, however, was unaffected by the addition of the TGF-β receptor 1 (TGFBR1) kinase inhibitor SB-431542 (Figure 1b). C1orf106 protein expression was also induced by TGF-β in a range of cancer and immortalised cell lines, including hepatocellular carcinoma cells (HepG2) (Figure 1c) and A549 lung adenocarcinoma cells (Figure 1d). Co-treatment of A549 cells with TGF-β and SB431542 blocked TGF-β mediated induction of C1orf106 (Figure 1d). In all tested cell lines, activation of the TGF-β pathway was indicated by p-SMAD2 (Figure 1b,c) or p-SMAD3 (Figure 1d). Basal C1orf106 expression in HepG2 and A549 was dependent on TGF-β, since reduced protein expression was observed upon receptor inhibition.

Expression of the murine homolog of *C1orf106,* 5730559C18Rik/*INAVA* (murine *C1orf106* hereafter) was significantly increased following TGF-β stimulation of murine embryonic and AKR-2B fibroblasts (Figure 1e,f). The ability of other ligands within the TGF-β superfamily to induce *C1orf106* mRNA expression was also assessed in human A431 epidermoid carcinoma cells. Of the eight cytokines tested, which included Activin and BMPs, only Activin-A and TGF-β could induce *C1orf106* mRNA to comparable levels (Appendix A).

### 3.2. C1orf106 Induction by TGF-β Is SMAD3-Dependent

SMAD2, 3 and 4 are key canonical signalling transducers of the TGF-β pathway, acting as transcription factors to regulate target gene expression. Therefore, we investigated the requirement for SMADs in regulating *C1orf106* induction downstream of TGF-β receptor activation. SMAD knockout mouse embryonic fibroblasts (MEFs) were stimulated with TGF-β and murine *C1orf106* expression determined by qRT-PCR (Figure 2a). While in wild-type MEFs, there was a strong induction of *C1orf106* upon TGF-β treatment, surprisingly, in the absence of SMAD2 and SMAD4 alone, murine *C1orf106* was still TGF-β inducible (although not significant for SMAD4; *p* = 0.0518). SMAD3 null MEFs had increased basal expression levels, and there was no TGF-β-mediated induction of *C1orf106* (Figure 2a). To test for any redundancy within the pathway that may account for these observations, SMAD2 null MEFs were transfected with siRNAs specific for *Smad3* or *Smad4* such that the expression of only one of the three SMADs remained intact (Figure 2b). In support of the data observed in SMAD3 null MEFs, murine *C1orf106* could not be induced by TGF-β following SMAD3 knockdown in SMAD2 −/− MEFs, whereas combined loss of SMAD2 and SMAD4 lowered baseline expression but did not impair induction (Figure 2c). *C1orf106* therefore appears to be induced by TGF-β stimulation by a previously undescribed mechanism whereby SMAD3 alone is necessary for induction. Interrogation of a recently published SMAD2/3 ChIP-seq dataset (GSE83788) identified significant enrichment of SMAD binding within intron 1 of the *C1orf106* gene locus in human breast cancer MII cells following TGF-β stimulation, suggesting it is a site of transcriptional regulation (Appendix A). Nucleotide alignment confirmed the conservation of this region and two potential SMAD binding elements (CAGAC) in human and murine *C1orf106* intron 1, further supporting its role as a SMAD-regulated region (Appendix A).

### 3.3. C1orf106 Expression Increases with Tumourigenic or Metastatic Potential in Breast Cancer Cell Line Models and Is a Mediator of Pro-Migratory and Pro-Invasive TGF-β Signalling

Having observed TGF-β-induced C1orf106 expression in a number of cancer cell types we sought to investigate its relevance in breast cancer. Comparison of a series of human MCF10A1 mammary cell lines [17] that included the MI (MCF10A1 spontaneously immortalised cell line from non-malignant human breast epithelium), MII (oncogenically initiated pre-malignant epithelium, oncogenic HRAS-transformed MI cells), MIII (line derived from a xenograft of MII cells that progressed to carcinoma and form predominantly well-differentiated carcinomas in nude mice) and the MIV cell line (line derived from a xenograft of MII cells that progressed to carcinoma that forms poorly differentiated carcinomas and is metastatic to the lung in tail-vein injections in nude mice) indicated an increase in C1orf106 basal protein expression levels (Figure 3a and Appendix A) and mRNA levels (Appendix A) in the carcinoma-derived cell lines MIII and MIV when compared to the premalignant cell lines, which was further increased by TGF-β treatment (Figure 3a and Appendix A), suggesting C1orf106 may have a role in progression to carcinoma. To investigate this further, we measured C1orf106 levels in the murine breast cancer progression series of sibling cell lines derived from a Balb/c mouse tumour [29], including the 67NR (tumourigenic but does not intravasate into blood vessels), 4T07 (metastatic to lungs but does not form metastatic outgrowths) and 4T1 cell lines (fully metastatic). We observed a stepwise increase in *C1orf106* mRNA levels (Figure 3c) and an increase in basal C1orf106 protein levels in both the 4T07 and 4T1 cell lines compared to the 67NR cell line (Figure 3d,e). TGF-β treatment readily induced C1orf106 protein expression in all three cell lines (Figure 3d,e). These results indicate that C1orf106 levels increase with attributes of tumourigenicity in the human cell lines and increased metastatic potential in the murine cell lines and that, when expressed, TGF-β signalling can further enhance C1orf106 expression.

As TGF-β promotes cancer cell migration, we investigated the potential involvement of C1orf106 in this response. In the murine model system, 4T1-luc cells (parental 4T1 cells stably expressing firefly luciferase) were lentivirally transduced with two independent hairpins targeting murine *C1orf106* or with a non-silencing control (NS) shRNA (Figure 3e). Transduction with shRNA 1 had little effect; however, efficient knockdown of C1orf106 was achieved using shRNA 2, and these cells were subsequently analysed in a scratch wound assay. C1orf106 knockdown had no impact on the rapid rate of basal migration; however, it did impact the ability of TGF-β to further promote the migratory ability of 4T1-luc cells (Figure 3f,g).

### 3.4. C1orf106 Enhances Clonogenicity In Vitro and Tumour Initiation Frequency In Vivo

The ability to migrate and invade are key contributors to the development of metastasis and clonogenicity, and self-renewal may also play a role in the ability of cells to seed tumours at primary and metastatic sites. The impact of C1orf106 expression levels was investigated in several assays of colony formation using 4T1-luc cells. Stable knockdown of murine *C1orf106* (Figure 4a) led to a slight reduction in the mammosphere-forming efficiency of 4T1-luc cells, a measure of anchorage-independent colony formation (Figure 4b), and in adherent colony-forming capacity (Figure 4c). However, in more stringent clonogenicity assays measuring self-renewal, where 4T1-luc cells were seeded at only 1 cell per well in a 96-well plate format, *C1orf106* knockdown profoundly inhibited the ability of single cells to establish a colony (Figure 4d). These assays suggested a requirement for basal C1orf106 in self-renewal and colony formation potential of 4T1 cells, particularly in limiting dilutions of cell number.

We next assessed whether increased C1orf106 expression could enhance tumourigenicity of 67NR cells—the less aggressive mouse mammary tumour cell line. 67NR cells were engineered to stably express either the empty vector pCDNA4/TO, pCDNA4/TO-C1orf106-GFP or pCDNA4/TO-C1orf106-Flag tagged constructs (Figure 5a). In a mammosphere assay, overexpression of C1orf106 was sufficient to significantly promote anchorage-independent colony formation of 67NR cells (Figure 5b). Additionally, in a low-density colony formation assay in adherent conditions, the gain of C1orf106 expression in 67NR cells enhanced colony formation (Figure 5c). Again, in the most stringent clonogenicity assay, C1orf106 expression significantly enhanced the clonogenic potential of 67NR cells (Figure 5d).

C1orf106 contains a domain of unknown function (DUF3338) at its N-terminus as determined by BLAST analysis. When this region was deleted, the resulting truncated protein ΔN(2–131)-C1orf106 had reduced capacity to promote 67NR clonogenicity when compared to overexpression of wild-type protein, suggesting that the DUF3338 domain may be critical for this function (Figure 5e,f). The effect of C1orf106 tumour initiation was then assessed in vivo. Using a limited dilution assay, orthotopic transplantation of 100 or 50 67NR cells resulted in the same percentage of recipient mice developing tumours irrespective of C1orf106 expression (Appendix A). However, when transplantation of only 10 cells to the mammary fat pad was carried out, 67NR cells with ectopic C1orf106 expression had an increased propensity to yield tumours compared to EV control 67NR cells (8/10 vs. 5/10; Appendix A), highlighting a tendency to enhance tumour initiation (Figure 5g; log-rank test: *p* = 0.2763).

### 3.5. High C1orf106 Expression Is Associated with Poor Clinical Outcome in Breast Cancer Patients

In vitro data indicated that C1orf106 levels increase with tumourigenicity/aggressiveness in cell line models. To investigate this correlation, *C1orf106/INAVA* expression in patients with distinct molecular subtypes of breast cancer was analysed using the bc-GenExMiner database. In keeping with our cell line analysis, patients with more aggressive tumours (basal-like and triple-negative) (n = 204) expressed significantly higher *C1orf106* mRNA than non-basal and not triple-negative tumours (TNBC) (n = 2370) (Figure 6a). Oncoprint analysis (www.cbioportal.org; accessed on 13 April 2023) showed amplifications being the most predominant genetic alteration of *C1orf106* (Figure 6b). Furthermore, through interrogation of the KMPlot database, high *C1orf106* expression correlated with a significantly shorter overall and post-progression-free survival, as well as an increased risk of metastatic relapse (Figure 6c). Altogether, we identified a role for the uncharacterised protein C1orf106 in promoting breast cancer cell line tumourigenicity, and high expression was positively correlated with poor prognosis of breast cancer patients, supporting a role for C1orf106 as a tumour promoter in breast cancer.

## 4. Discussion

We have identified *C1orf106* as a novel SMAD3-dependent TGF-β target gene in a variety of immortalised and cancer cell types (Figure 1). Our data are supported by publicly accessible microarray datasets, which show that *C1orf106* is induced 2.8-fold following TGF-β treatment of HaCaT cells [30]. The additional canonical TGF-β signalling mediators SMAD2 and SMAD4 were dispensable for C1orf106 induction, deviating from the described TGF-β signal transduction pathway to gene regulation (Figure 2). SMAD4 independent TGF-β target gene regulation has been described previously [31]; however, SMAD3 dependence as a lone mediator of TGF-β transcriptional responses has not yet been explicitly reported, and the molecular mechanism underlying our observation is not yet understood. It is possible that SMAD3 may bind in collaboration with SMAD2 and SMAD4 at the *C1orf106* gene locus, but their presence has no impact on TGF-β-mediated transcriptional regulation. Alternatively, SMAD3 interaction with competing transcription factors may disengage SMAD–SMAD interactions and direct SMAD3 to the *C1orf106* gene locus independently of SMAD2 and SMAD4. Such a mechanism has been described in relation to SMAD4 independency involving the competing transcription factors TIF-1γ [32] and TTF-1γ [33], which bind SMAD2/3; however, redundancy between SMAD2 and SMAD3 was not investigated in these studies. A potential region of SMAD3 binding within an intron of *C1orf106* was identified in a SMAD2/3 ChIP-sequencing analysis of TGF-β target genes in MII breast cancer cells (Appendix A) [34]. Further validation of this as a TGF-β responsive element will help elucidate the molecular basis of SMAD3-regulation of *C1orf106* expression and identify potential co-factors.

As well as the identification of *C1orf106* as a TGF-β target gene, we found that *C1orf106* mRNA and protein expression increases with tumourigenicity or metastatic potential in breast cancer cell line models and found a significant correlation of high *C1orf106* expression with poor clinical outcomes and disease aggressiveness in human data sets (Figure 3, Figure 4, Figure 5 and Figure 6). In keeping with this, our experimental data indicate a pro-tumourigenic function for C1orf106 in breast cancer cell lines, suggesting an important role in the transition from premalignant (MII) to carcinoma (MIII/MIV) (Figure 3a). In the murine progression series, the less tumourigenic 67NR cells have been described as having a luminal subtype owing to nuclear ERα positivity, while the more aggressive 4T1 sibling cell line is regarded as basal-like [35]. Our *C1orf106* expression analysis in these lines corroborates findings from gene expression analysis of human breast cancer tumours, indicating that C1orf106 is expressed at higher levels in more aggressive basal and triple-negative subtypes (Figure 3). This finding is in line with a recent study showing that *C1orf106* is amplified in aggressive basal-like breast cancer, maintaining the luminal progenitor gene expression profile by regulating ELF5 and GATA3 expression [25].

In line with our observations that C1orf106 promotes TGF-β-induced migration of murine breast cancer cells (Figure 3f,g), C1orf106 has also been shown to promote migration, invasion and metastasis in papillary thyroid cancer, which is proposed to occur via an FGF1-MMP9 axis [36]. MMP9 has been shown to contribute to TGF-β-induced invasion in MII cells [37], which may be relevant to the mechanism of action of C1orf106 in our breast cancer model. Interestingly, an opposing phenotype has been observed in the gut epithelium, such that the loss of C1orf106 expression enhanced migration of intestinal epithelial cells [24]. It is possible that C1orf106 exerts differential functions dependent on cell and tissue type, similar to its master regulator TGF-β, and its role in promoting epithelial integrity may in fact act as a barrier to colorectal tumourigenesis.

As well as having a role in potentiating migration and invasion, we found that basal C1orf106 expression is required for self-renewal of breast cancer cells. Anchorage-independent mammosphere formation and anchorage-dependent clonogenic survival were promoted by C1orf106 expression, as well as the ability to seed tumours at extremely low density (Figure 4 and Figure 5g). The potential of C1orf106 to drive colony formation is in line with another study in breast cancer cell lines [25]. Enhanced clonogenicity was dependent on the DUF3338 domain which C1orf106 shares with FRMD4a, FRMD4b and CCDC120 (Figure 5a–f). FRMD4a has been linked to maintenance of epithelial polarity through interactions of the DUF3338 domain with cytohesins [38] and CCDC120 was recently implicated in centrosome-microtubule anchoring [39]. Further, the DUF3338 domain of C1orf106 binds ARNO, a cytohesin ARF-GEF, and thus maintains the epithelial barrier function [40]. Given the involvement of the DUF3338 region, the enhanced self-renewal ability could occur as a result of changes in epithelial polarity/barrier function or centrosome positioning, both of which are determinants in maintaining stem cell populations by segregation of cell fate factors. To date, the results so far suggest that the pro-migratory functions of C1orf106 may be context or transformed cell-type specific, being restricted to breast and thyroid cancer cells. Further, promotion of clonogenicity and migration by C1orf106 are important determinants of tumour cells’ metastatic ability to disseminate to and colonise secondary sites. Taken together, SMAD3-dependent increased C1orf106 expression may endow cells with the increased capacity to metastasize through enhanced TGF-β-induced migration, anchorage-independent growth and clonogenic survival for which the DUF3338 domain appears to be crucial. Further, our study underlines the potential role of C1orf106 as a biomarker for breast cancer [25] and a putative target downstream of TGF-β signalling.

## Figures and Tables

**Figure 1 cells-13-01530-f001:**
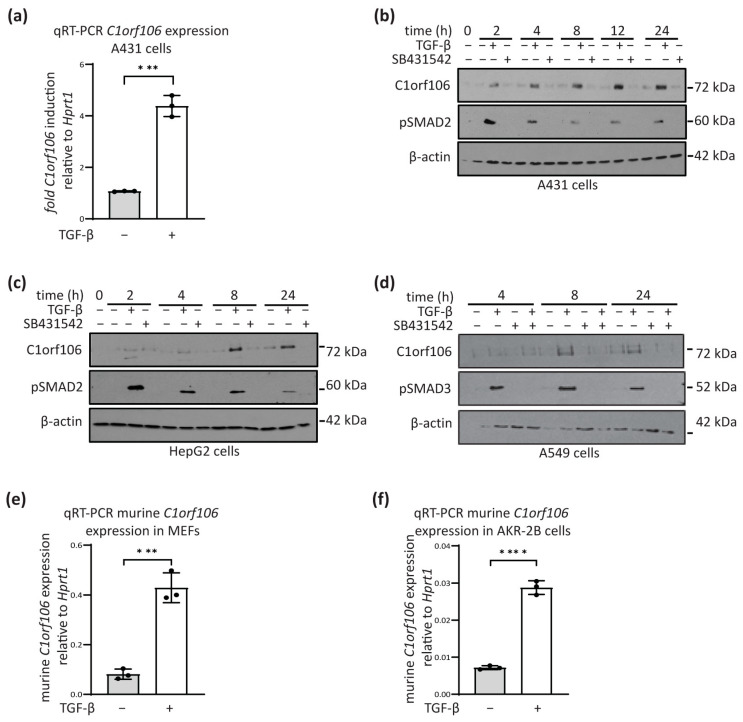
*C1orf106* is a novel and robust transcriptional TGF-β target gene in human and murine cell lines. (**a**) *C1orf106* mRNA expression in A431 cells following 2 h stimulation with 5 ng/mL TGF-β or vehicle control was analysed by qRT-PCR (n = 3). *C1orf106* expression was normalized to *Hprt1*. (**b**–**d**) C1orf106 protein expression in cells following 5 ng/mL TGF-β pathway stimulation for the indicated times or inhibition with 10 μM SB431542 was analysed by Western blotting. β-actin was used as a loading control. (**e**,**f**) Murine *C1orf106* mRNA expression following 5 ng/mL TGF-β stimulation for 2 h was analysed by qRT-PCR in the indicated cell lines (n = 3). Murine *C1orf106* Ct values were normalised to *Hprt1*. Statistical analysis was carried out by unpaired Students *t*-test. *** = *p* < 0.001; **** = *p* < 0.0001.

**Figure 2 cells-13-01530-f002:**
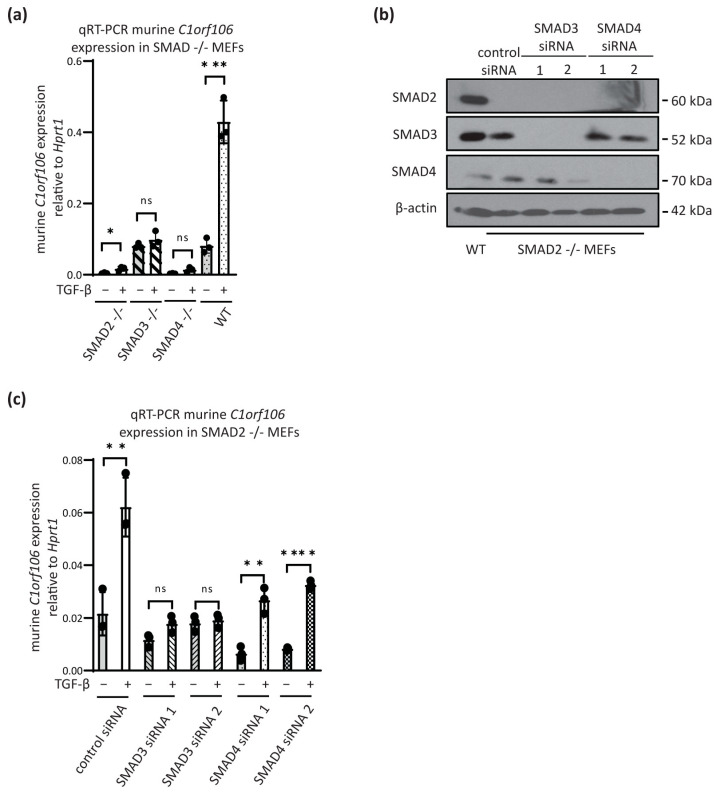
*C1orf106* transcriptional induction is SMAD3-dependent. (**a**) Comparison of murine *C1orf106* mRNA expression in SMAD wild type (WT) and SMAD knockout mouse embryonic fibroblasts (MEFs) stimulated with 5 ng/mL TGF-β for 2 h. Murine *C1orf106* Ct values were normalized to *Hprt1* (n = 3 biological replicates). The data points for wild-type MEFs stimulated with TGF-β are the same as in Figure 1e. (**b**) Western blot analysis of SMAD2, SMAD3 and SMAD4 protein expression following siRNA of *Smad3* and *Smad4*. β-actin was used as a loading control, and SMAD2 wildtype MEFs were used as a positive control. (**c**) qRT-PCR analysis of murine *C1orf106* expression in response to TGF-β (5 ng/mL for 2 h) stimulation in SMAD2 −/− MEFs following siRNA knockdown of *Smad3* and *Smad4* (n = 3 biological replicates). Error bars show standard deviation. Statistical significance was determined by unpaired student’s *t*-test. * = *p* < 0.05; ** = *p* < 0.01; *** = *p* < 0.001; **** = *p* < 0.0001; ns = non-significant.

**Figure 3 cells-13-01530-f003:**
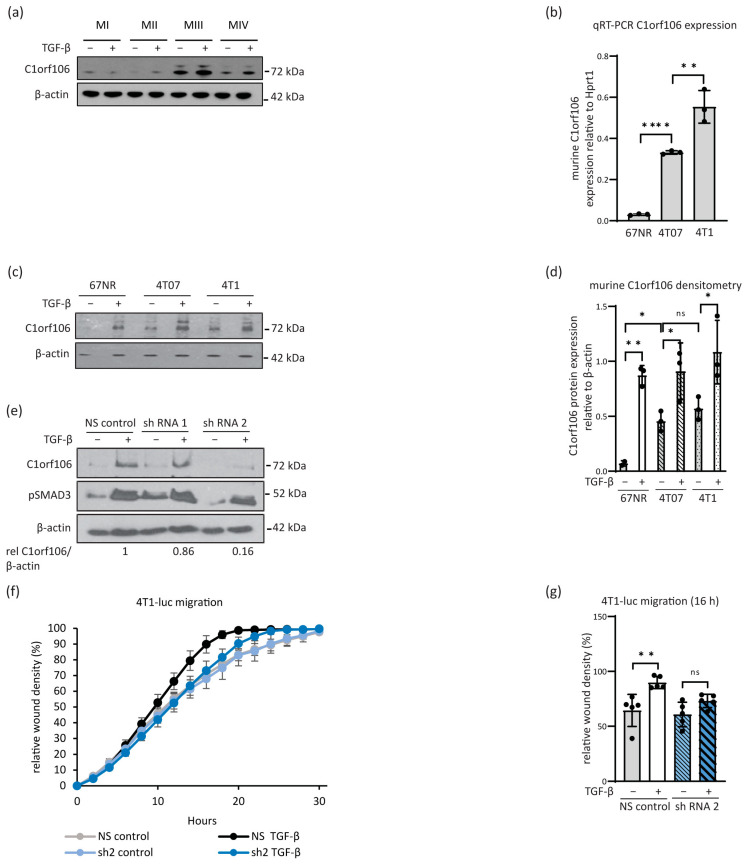
*C1orf106* expression increases with disease progression in experimental models of breast cancer and is a mediator of pro-migratory TGF-β signalling. (**a**) Comparison of C1orf106 protein expression in MI, MII, MIII and MIV breast cancer cells stimulated with 5 ng/mL TGF-β for 8 h by Western blotting. β-actin was used as a loading control. (**b**) Murine *C1orf106* mRNA expression in cells from the 4T1 progression series of breast cancer (n = 3 biological replicates). Murine *C1orf106* Ct values were normalised to *Hprt1*. (**c**,**d**) C1orf106 protein expression in indicated cell lines treated with 5 ng/mL TGF-β for 4 h was analysed by Western blotting ((**c**), representative blot is shown) and was quantified by densitometry in ImageJ relative to the β-actin sample integrity control ((**d**), n = 3 biological replicates). (**e**) 4T1-luc cells were lentivirally transduced with a non-silencing control shRNA or two independent hairpins targeting murine *C1orf106*, treated with 5 ng/mL TGF-β for 4 h and analysed by Western blotting. β-actin was used as a loading control. Relative C1orf106 expression to β-actin has been calculated by densitometry (values shown below). (**f**,**g**) Scratch wound migration of 4T1-luc *C1orf106* knockdown cells in response to stimulation with 5 ng/mL TGF-β. Cells were imaged every two hours using the IncuCyte™ Zoom platform (Essen Bioscience, Royston, UK) post-wounding, and per cent confluence of the wound was used as a readout of migration (n = 5 technical replicates in one experiment). Relative wound density at 16 h (**g**). Error bars show standard deviation. Statistical analysis was performed by unpaired student’s *t*-test. * = *p* < 0.1; ** = *p* < 0.01; **** = *p* < 0.0001; ns = non-significant.

**Figure 4 cells-13-01530-f004:**
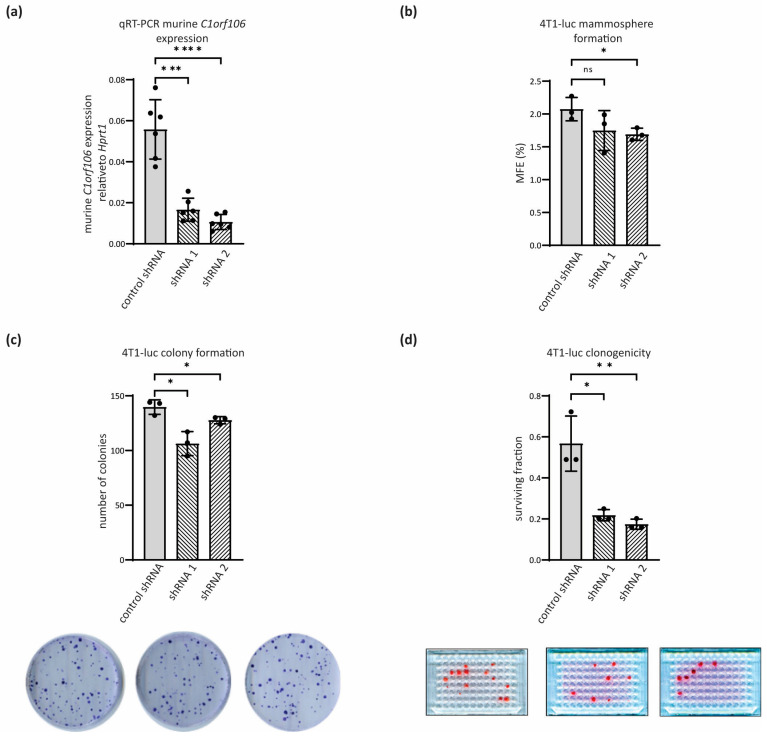
Loss of *C1orf106* impairs clonogenic potential in metastatic murine mammary cells. (**a**) Murine *C1orf106* mRNA expression in 4T1-luc *C1orf106* knockdown cells was determined by qRT-PCR. Murine *C1orf106* Ct values were normalised to *Hprt1* (n = 3 biological replicates with two technical replicates each). (**b**) Mammosphere forming efficiency ((number of mammospheres formed/number of cells seeded) × 100) of 4T1-luc *C1orf106* knockdown cells. Colonies > 50 μm were counted 5 days post-seeding in anchorage-independent conditions (n = 3 biological replicates). (**c**) Colony formation of 4T1-luc *C1orf106* knockdown cells 7 days post-seeding of 200 cells/10 cm dish. Colonies were stained with borax:toluidine blue, and colonies > 50 cells were counted. Representative plates from each cell line are shown (n = 3 biological replicates). (**d**) 4T1-luc *C1orf106* knockdown cells were seeded at 1 cell/well in a 96-well plate, and colonies were stained with sulphorhodamine b after one week (n = 3 biological replicates). Colonies > 50 cells were counted, and the surviving fraction was calculated. Representative plates from each cell line are shown. Error bars show standard deviation. Statistical analysis was performed by unpaired student’s *t*-test. * = *p* < 0.1; ** = *p* < 0.01; *** = *p* < 0.001; **** = *p* < 0.0001; ns = non-significant.

**Figure 5 cells-13-01530-f005:**
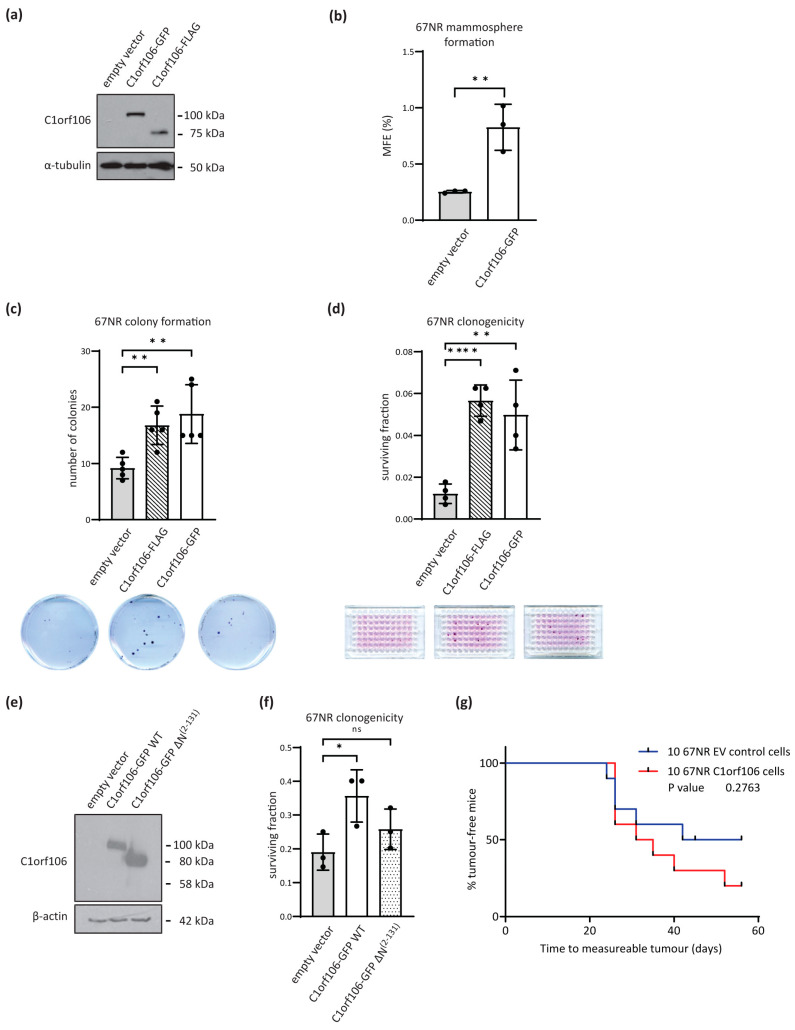
Increased *C1orf106* in non-metastatic breast cancer cells promotes clonogenicity and enhances tumour initiation capacity in vivo. (**a**) Western blot analysis of C1orf106 protein levels in 67NR cells transfected with pCDNA4/TO empty vector, pcDNA4/TO C1orf106-GFP or pcDNA4/TO C1orf106-Flag following zeocin selection. α-tubulin was used as a loading control. (**b**) Mammosphere-forming efficiency of 67NR empty vector and C1orf106 overexpressing cells. Colonies > 50 μm were counted after 5 days in culture (n = 3 biological replicates). (**c**) 67NR empty vector and C1orf106 overexpressing cells were seeded at 100 cells/10cm dish to assess colony-forming capacity (n = 5 biological replicates). Cells were stained after 14 days in culture and colonies > 50 cells counted. Representative plates from each cell line are shown. (**d**) 67NR empty vector and C1orf106 overexpressing cells were seeded at 1 cell/well in 96-well plates and cells stained with sulphorhodamine B after 14 days (n = 4 biological replicates). Colonies > 50 cells were counted, and the surviving fraction was calculated. Representative plates from each cell line are shown. (**e**) 67NR cells were transfected with pcDNA4/TO empty vector, pcDNA4/TO C1orf106-GFP wildtype or pCDNA4/TO C1orf106-GFP ΔN(2–131) and protein expression was analysed by Western blotting. β-actin was used as a sample integrity control. (**f**) 67NR cells were assessed for clonogenicity as in (**d**) (n = 3 biological replicates). (**g**) 67NR empty vector and C1orf106-GFP cells were orthotopically transplanted into the mammary fat pad of female CD-1 mice. Tumours were monitored by calliper measurement thrice weekly, and time to measurable tumour initiation is shown (n = 10 mice). Statistical significance was determined by unpaired student’s *t*-test (**b**–**d**,**f**) and log-rank survival analysis (**g**). * = *p* < 0.1; ** = *p* < 0.01; **** = *p* < 0.0001; ns = non-significant.

**Figure 6 cells-13-01530-f006:**
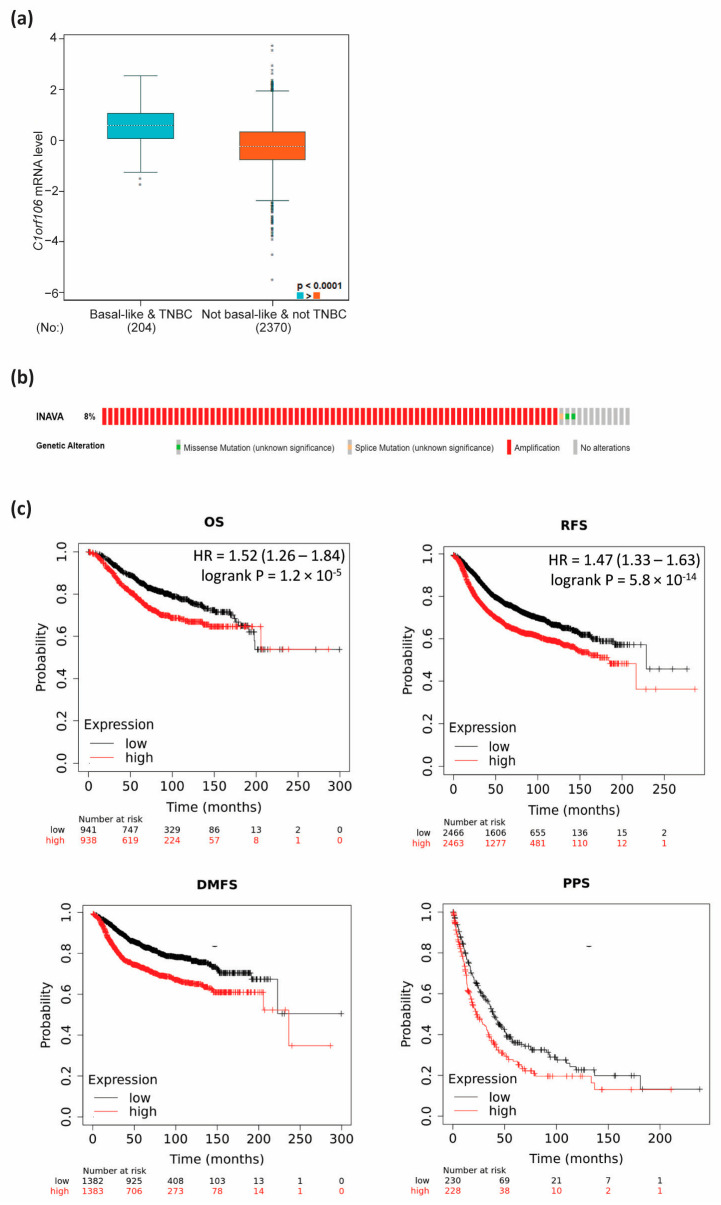
High *C1orf106* mRNA expression is associated with poor survival outcomes in breast cancer patients. (**a**) bc-GenExMiner analysis of *C1orf106* mRNA (Probe ID 209010) expression according to molecular subtype. (**b**) Oncoprint analysis (www.cbioportal.org) of invasive breast carcinoma samples from TCGA PanCancer Atlas including 996 patients/samples shows genetic alterations (mainly amplifications) of *C1orf106/INAVA* in 8% of the samples. (**c**) Kaplan–Meier survival analysis across indicated clinical correlates of breast cancer patients split based on median *C1orf106* mRNA expression. Data were generated using the KMPlot tool. OS = overall survival; RFS = relapse-free survival; DMFS = distant-metastasis-free survival; PPS = post-progression-free survival.

## Data Availability

All data supporting the findings of this study are available in the paper and its Appendix A section. Further enquiries can be directed to the corresponding author.

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
