# Peer review of "C1orf106 (INAVA) Is a SMAD3-Dependent TGF-β Target Gene That Promotes Clonogenicity and Correlates with Poor Prognosis in Breast Cancer"

_cells, 2024, doi:10.3390/cells13181530_

Round 1

Reviewer 1 Report

Comments and Suggestions for Authors

The authors show that TGF-beta induces C1orf106 in several tumour cell lines, and that this is connected with enhanced migration and invasion. Moreover, high expression of C1orf106 mRNA correlates with aggressiveness and poor prognosis in human breast cancer. These are interesting findings which deserve to be published after some revision.

Specific points:

1. The authors claim that the TGF-beta-induction of C1orf106 is dependent on SMAD3, but not on SMAD2 and SMAD4 (Fig. 2). However, it is clear from Fig, 1a that knock-out or either of the SMADs suppresses C1orf106 expression. A small induction is seen by TGF-beta stimulation of SMAD2 knock-out MEFs (Fig. 2c), which is lost if SMAD3 is knocked down. However, also knock-down of SMAD4 has an effect, although some TGF-beta inducibility remains. Thus, the dependency of different SMAD isoforms is not so straight-forward. To claim that the induction is SMAD4-independent, they need to do further experiments, e.g. using the SMAD4 knock-out MEFs. If further evidence is found for SMAD4-independence, it would be interesting to know if knock-down of TIF-1gamma has any effect.

2. Fig. 3: The authors claim that C1orf106 expression correlates with the degree of malignancy, however, there appears not to be any difference in protein expression between MII and MIV cells (Fig. 3b).

3. Fig. 6a: This panel needs improvement in its layout; it is not possible to read the text. Moreover, the reference to this figure in the text is unclear: "patients with more aggressive tumours (basal-like and triple negative) (n=204) expressed significantly higher c1orf106 mRNA than non-basal and triple negative tumours...". "Triple negative comes twice! What is meant?

4. There seems to have been a shift in the numbers to the references in the Discussion. 

Author Response

Reviewer comment: The authors show that TGF-beta induces C1orf106 in several tumour cell lines, and that this is connected with enhanced migration and invasion. Moreover, high expression of C1orf106 mRNA correlates with aggressiveness and poor prognosis in human breast cancer. These are interesting findings which deserve to be published after some revision.

Response: We thank the reviewer for the thorough review of our manuscript and are pleased that they recommend publication following revision. We address each of their specific points below.

Reviewer Specific points:

  1. The authors claim that the TGF-beta-induction of C1orf106 is dependent on SMAD3, but not on SMAD2 and SMAD4 (Fig. 2). However, it is clear from Fig, 1a that knock-out or either of the SMADs suppresses C1orf106 expression. A small induction is seen by TGF-beta stimulation of SMAD2 knock-out MEFs (Fig. 2c), which is lost if SMAD3 is knocked down. However, also knock-down of SMAD4 has an effect, although some TGF-beta inducibility remains. Thus, the dependency of different SMAD isoforms is not so straight-forward. To claim that the induction is SMAD4-independent, they need to do further experiments, e.g. using the SMAD4 knock-out MEFs. If further evidence is found for SMAD4-independence, it would be interesting to know if knock-down of TIF-1gamma has any effect.

Response: We investigated the dependence of TGF-β mediated induction of C1orf106 by utilising WT, SMAD2, SMAD3 and SMAD4 knockout MEFs. Our qRT-PCR results indicate that C1orf106 expression is induced by TGF-β stimulation in WT and SMAD2 ko MEFs and to a small extent in SMAD4 KO mefs (Figure 2a) although this just failed to reach statistical significance (p=0.0518). This data clearly indicated that SMAD2 is not required for TGF-β signalling to induce expression of c1orf106 and suggested that SMAD3 could be required and that SMAD4 may also be dispensable for this induction. To further investigate this, we efficiently knocked down expression of SMAD3 and SMAD4 with two different siRNAs for each gene in SMAD2 null mefs. Knockdown of SMAD3 totally abrogated expression of SMAD3 whilst leaving SMAD4 expression intact and similarly knock down of SMAD4 completely abrogated expression of SMAD4 whilst leaving expression of SMAD3 intact (Figure 2b). Knockdown of SMAD3 blocked the ability of TGF-β to induce C1orf106 whereas knockdown of SMAD4 did not abrogate the TGF-β mediated induction of C1orf106. We believe this data clearly indicates that SMAD3 is required for TGF-β to induce C1orf106 and that SMAD4 is not and hope that the reviewer will agree with this interpretation. It is interesting that knockdown of SMAD4 affected the baseline level of expression of C1orf106 as the reviewer has pointed out and we agree it would be interesting to see if knockdown of TIF1-gamma has any effect in the future but we believe this is beyond the scope of the current study. The potential role of TIF1γ is mentioned in the discussion. We have amended the text as follows (additions in italics)

Line 195:-“combined loss of SMAD2 and SMAD4 lowered baseline expression but did not impair induction (Figure 2c).”  

Line 196:- “induced by TGF-β stimulation by a previously undescribed mechanism whereby SMAD2 and SMAD4 are dispensable and SMAD3 alone is necessary for induction”.

Reviewer point: 2. Fig. 3: The authors claim that C1orf106 expression correlates with the degree of malignancy, however, there appears not to be any difference in protein expression between MII and MIV cells (Fig. 3b).

Response: We have now expanded this analysis to include the MI (MCF10A1 spontaneously immortalised cell line from non-malignant human breast epithelium ), MII (oncogenically initiated pre-malignant epithelium, oncogenic HRAS transformed MI cells), MIII (line derived from a xenograft of MII cells that progressed to carcinoma and form predominantly well-differentiated carcinomas in nude mice) and the MIV cell lines (line derived from a xenograft of MII cells that progressed to carcinoma that forms poorly differentiated carcinomas and is metastatic to the lung in tail vein-injections in nude mice). Western blot analysis indicates a marked increase in basal C1orf106 expression between the MII-MIII transition (new Figure 3a,new Figure A3a,b).  MIV cells express more C1orf106 protein and mRNA than MII cells but intriguingly less than MIII cells likely reflecting their different origins and a potential threshold effect of C1orf106 expression. We have re-written the corresponding section of the manuscript, abstract and discussion to reflect these findings.

Reviewer point 3. Fig. 6a: This panel needs improvement in its layout; it is not possible to read the text. Moreover, the reference to this figure in the text is unclear: "patients with more aggressive tumours (basal-like and triple negative) (n=204) expressed significantly higher c1orf106 mRNA than non-basal and triple negative tumours...". "Triple negative comes twice! What is meant?

Response: We apologise for the lack of clarity in the figure and the error in the accompanying text. We have improved the labelling and corrected the error in the text to now read “patients with more aggressive tumours (basal-like and triple negative) (n=204) expressed significantly higher c1orf106 mRNA than non-basal and not triple negative tumours.”

Reviewer point 4. There seems to have been a shift in the numbers to the references in the Discussion. 

Response: We thank the reviewer for pointing out this error which has now been corrected. 

Reviewer 2 Report

Comments and Suggestions for Authors

Strathearn and colleagues identified a new target gene of TGF-β.

The work is very elegantly organised and the results are clearly and comprehensively explained.

A very good paper.

Author Response

Reviewer Comment: Strathearn and colleagues identified a new target gene of TGF-β.

The work is very elegantly organised and the results are clearly and comprehensively explained.

A very good paper.

Response: We thank the reviewer for their appreciation of our paper.

Reviewer 3 Report

Comments and Suggestions for Authors

In the manuscript entitled: "C1orf106 (INAVA) is a SMAD3-dependent TGF-β target gene that promotes clonogenicity and correlates with poor prognosis in breast cancer," Lauren S. Strathearn and colleagues define C1orf106 as a novel TGF-β target gene induced in an SMAD3-dependent manner in human and murine cell lines. They also showed that C1orf106 expression positively correlates with disease progression in human and murine breast cancer cell lines and is required for enhanced migration and invasion after TGF-β stimulation.

The study is correctly designed and technically rational. The analyses are performed appropriately.

Below is a list of some points to address:

Figure 1b. The authors state that "C1orf106 protein levels and this induction was blocked by co-treatment with the TGFBR1 kinase inhibitor SB-431542," but in Figure 1b, the co-treatment with TGF-β plus SB-431542 is not indicated.

Figure 3g-h Representative images of scratch wound migration of 4T1-luc C1orf106 knockdown cells in response to stimulation with 5 ng/ml TGF-β would be required.

Figure 4d. Colonies stained with sulphorhodamine b. It is challenging to count colonies of 50 cells in a 96-well plate. The authors should read the corresponding absorbance of the SRB assay and add it to the figure.

Figure 5d: See the previous comment.

Figure 6 and b. The figures must be labeled with legible writing.

There are some typing errors to fix.

Comments on the Quality of English Language

The quality of the English language is good.

There are some typing errors to fix.

Author Response

Reviewer Comments: In the manuscript entitled: "C1orf106 (INAVA) is a SMAD3-dependent TGF-β target gene that promotes clonogenicity and correlates with poor prognosis in breast cancer," Lauren S. Strathearn and colleagues define C1orf106 as a novel TGF-β target gene induced in an SMAD3-dependent manner in human and murine cell lines. They also showed that C1orf106 expression positively correlates with disease progression in human and murine breast cancer cell lines and is required for enhanced migration and invasion after TGF-β stimulation.

The study is correctly designed and technically rational. The analyses are performed appropriately.

Response: We thank the reviewer for their careful review of our manuscript and their positive comments. We address the points they raise below to the best of our ability and hope that they find them acceptable.

Reviewer points: Below is a list of some points to address:

Figure 1b. The authors state that "C1orf106 protein levels and this induction was blocked by co-treatment with the TGFBR1 kinase inhibitor SB-431542," but in Figure 1b, the co-treatment with TGF-β plus SB-431542 is not indicated.

Response: We thank the reviewer for pointing out our error in the text which has now been corrected. The co-treatment statement refers to Figure 1d in A549 cells.

New text “Increased expression of C1orf106 protein following TGF-β treatment of A431 cells was confirmed by Western blotting (Figure 1b). Basal expression in this cell line, however, was unaffected by the addition of the TGF-β receptor 1 (TGFBR1) kinase inhibitor SB-431542 (Figure 1b). C1orf106 protein expression was also induced by TGF-β in a range of cancer and immortalised cell lines including hepatocellular carcinoma cells (HepG2) (Figure 1c) and A549 lung adenocarcinoma cells (Figure 1d). Co-treatment of A549 cells with TGF-β and SB431542 blocked TGF-β mediated induction of C1orf106 (Figure 1d)”.

Reviewer point: Figure 3g-h Representative images of scratch wound migration of 4T1-luc C1orf106 knockdown cells in response to stimulation with 5 ng/ml TGF-β would be required.

Response: We agree with the reviewer that representative images would be nice to show. However, these assays were performed in Dundee and the raw image archive was deleted when we moved to Glasgow. The graphical analysis of the relative wound density over time calculated with the incucyte software is presented in Figure 3g and statistical analysis of the data at the 16 hour time point is presented in Figure 3h and we hope that the reviewer will please find this sufficient to illustrate that knockdown of C1orf106 impairs the ability of TGF-β to stimulate cell migration in these cells.  

Reviewer point: Figure 4d. Colonies stained with sulphorhodamine b. It is challenging to count colonies of 50 cells in a 96-well plate. The authors should read the corresponding absorbance of the SRB assay and add it to the figure. Figure 5d: See the previous comment.

Response: We routinely use microscopy to count cell colonies >50 cells in these assays and do not measure SRB absorbance. Changes in SRB can also reflect changes in cell size as well as cell number and not just colony number, so we prefer to use microscopy for counting cell colonies. For the assays shown in our figures we no longer have the original plates and so we cannot retrospectively calculate the absorbance readings. We hope that the reviewer will accept this explanation. 

Reviewer point: Figure 6 and b. The figures must be labeled with legible writing.

Response: We apologize for any lack of clarity, and we have increased the font size of the labels and hope that this is now acceptable.

Reviewer point: There are some typing errors to fix.

Response: We have performed a further proof of the manuscript and have addressed any typing errors.

Round 2

Reviewer 1 Report

Comments and Suggestions for Authors

The revision of this paper has been appropriate and the paper is now ready for publication. However, first a small change in the wording should be considered: On line 517-519 the authors say that "SMAD2 and SMAD4 are dispensible and SMAD3 alone is necessary for induction" of C1orf106. Given the fact that the expression was decreased about 10-fold after knock-down of either SMAD2 or SMAD4, "dispensible" is not an appropriate formulation. The authors should use a better formulation. 

Author Response

The revision of this paper has been appropriate and the paper is now ready for publication. However, first a small change in the wording should be considered: On line 517-519 the authors say that "SMAD2 and SMAD4 are dispensible and SMAD3 alone is necessary for induction" of C1orf106. Given the fact that the expression was decreased about 10-fold after knock-down of either SMAD2 or SMAD4, "dispensible" is not an appropriate formulation. The authors should use a better formulation. 

Response: We thank the reviewer for recommending publication. In regards to the use of the word dispensable we have now deleted the phrase "SMAD2 and SMAD4 are dispensable"

Reviewer 3 Report

Comments and Suggestions for Authors

The authors did not respond to all requests but overall they made satisfactory improvements to the work.

By the way, the use of SRB absorbance is commonly accepted for colony counting, which certainly also takes into account cell size, which is not insignificant in tumor development.

Comments on the Quality of English Language

English is fine. Checking for the minor errors

Author Response

The authors did not respond to all requests but overall they made satisfactory improvements to the work.

By the way, the use of SRB absorbance is commonly accepted for colony counting, which certainly also takes into account cell size, which is not insignificant in tumor development.

Response: We thank the reviewer for accepting that our improvements to the manuscript are satisfactory and for clarifying the utility of the SRB assay